# OpenReview forum: "Large Language Models are Capable of Offering Cognitive Reappraisal, if Guided"
_colmweb.org/COLM/2024/Conference — COLM_

### Official Review · Reviewer_ArmX · 2024-05-09

**Rating:** 6
**Confidence:** 3
**Ethics Flag:** 1

**Summary:**

Long-term mental well-being requires emotional self-regulation, starting by engaging with cognitive reappraisals that uses language to change negative appraisals that an individual makes of the situation. This paper hypothesizes that this task can be elicited from LLMs if they are guided by carefully crafted principles. Based on this, this paper introduces RESORT (REappraisals for emotional SuppORT), a psychologically-grounded framework that defines a constitution for a series of dimensions, motivated by the cognitive appraisal theories of emotions.

The authors conduct an extensive evaluation of LLMs (GPT4-v, Llama2 13 B and Mistral 7B) for their cognitive reappraisal capability by clinical psychologists with M.S. or Ph.D. degrees, who judged LLM outputs (as well as human responses) in terms of their alignment to psychological principles, perceived empathy, as well as any harmfulness or factuality issues. Experimental results show that LLMs (even those at the 7B scale) produce cognitive reappraisals that significantly outperform human-written responses as well as non-appraisal-based prompting.

**Questions To Authors:**

1.	I am interested in how can LLM response better than human oracle, written by PhD student in psychology. It will be better if the authors can show some case study, as well as the human annotation to provide some insight.
2.	Also, providing a case study about the responses with and without appr and cons may help the reader to understand the effectiveness of each component. I find a case study in Table 6 in the Appendix. It will be better to mention it in the main paper.
3.	The evaluation is too subjective and rely on expert knowledge, making it challenging to judge the effectiveness, even for the reviewer. For example, for the example in Figure 2, it is not very clear to me that how the guided response is better than unguided.
4.	The examples in Table 9 are also a little bit confusing. It is not clear to me why the first response is Lack of Specific Guidelines / Actionable Steps.

**Reasons To Accept:**

1.	This paper is well-written and easy to follow. All the implementation details are clearly shown in the main paper, as well as the appendix.
2.	The authors evaluate the performance by clinical psychologists with M.S. or PhD. Degrees, making the findings more sound.
3.	The proposed method can significantly outperform baseline methods and even oracle responses.

**Reasons To Reject:**

1.	The method is relatively too simple without too much technical contribution.
2.	The scope and the focus of this paper, Cognitive Reappraisal, is too narrow and specific, limited the potential impact of this paper.

---

> ### Author Rebuttal · Authors · 2024-05-30
>
> Thank you!
>
> > **Technical contribution:**
>
> As Reviewers WuPB & Zhu4 noted, we propose an innovative framework that induces advanced cognitive capabilities in LLMs, proving effective even at the 7B scale. Besides, as Reviewer LP97 highlighted, we introduce a novel benchmark for evaluating LLMs’ ability to generate cognitive reappraisals.
>
> > **Narrow focus of this paper:**
>
> Reappraisal is a specific and well-defined strategy for providing emotional support, backed by extensive psychological literature with broad applications. There is award-winning literature in NLP on providing reappraisals and “positive reframes” (Ziems et al., 2022; Sharma et al., 2020), so we respectfully disagree with the claim that this topic is “narrow”.
>
> Moreover, RESORT can guide response generation for different theoretical constructs with expert-crafted constitutions. We also demonstrate how to use experts (in our case psychologists) to validate the output. Consistent with prior work (Ayers et al., 2023; Yin, Jia, & Wakslak, PNAS 2024; Li, Herderich, & Goldenberg, 2024), our study underscores the potential of LLMs on providing emotional support.
>
> > **How are LLM responses better than human oracle ones?**
>
> The oracle responses were often perceived as informal, vague, and lacking detailed suggestions or support for the OP. Providing reappraisals is inherently cognitively-demanding, even for experienced psychologists. We hope AI can help “augment human ability” in such tasks, and this work is a first step in that direction.
>
> > **Case study on with/without appr and cons:**
>
> We will discuss Table 6 (Appendix D) in the main body of the paper.
>
> > **Subjective eval:**
>
> We agree that the eval is subjective by nature, hence we rely on expert psychologists as evaluators. Despite the subjectivity, they still reach moderate agreement as shown in Table 2 (Artstein & Poesio, 2008). Evaluators received clear instructions (Figures 7&8) to guide their assessment. For example, the “unguided response” in Figure 2 is unempathetic and lacks specific coping mechanisms, whereas the “guided response” identifies the root cause of distress and provides specific solutions (we will include the full response for clarity).
>
> > **First example in Table 9:**
>
> The suggestions in the response are mostly generic. For example, advice like “It might be helpful … as a whole” is templated and can be applied to any context. In our evaluation, we discourage responses that are not tailored to what the narrator is facing.

---

### Official Review · Reviewer_Zhu4 · 2024-05-09

**Rating:** 6
**Confidence:** 3
**Ethics Flag:** 1

**Summary:**

This paper explores the potential of Large Language Models (LLMs) to offer cognitive reappraisal, a psychological strategy that helps individuals change their negative appraisals of situations. The authors introduce the RESORT framework, which consists of reappraisal constitutions across multiple dimensions that can guide LLMs in generating empathic responses to support individuals in reappraising their situations. The study includes an expert evaluation by clinical psychologists, showing that LLMs guided by RESORT can produce effective cognitive reappraisal responses to social media messages seeking support.

**Reasons To Accept:**

1. The paper introduces a novel application of LLMs in providing cognitive reappraisal, showcasing the potential for advanced psychological capabilities of LLM.
2. They conducted an extensive evaluation of LLMs for their cognitive reappraisal capability.

**Reasons To Reject:**

1. In the experiment, Oracle responses were written by one person and may have subjective biases. In the experiment, it can be seen that the performance often deteriorates when +app+cons are used simultaneously.
2. LLM is sometimes sensitive to prompt words. Have you tested the performance of different Constructions prompt words.
3. Is there any other data source, such as real users asking questions when consulting with psychologists. Experiments from different data sources can demonstrate the generalization of the method.

---

> ### Author Rebuttal · Authors · 2024-05-30
>
> Thank you for your positive comments on the innovation of the RESORT framework to induce advanced cognitive capabilities from LLMs, which Reviewers WuPB and LP97 also pointed out. We are also grateful for your appreciation of our rigorous and in-depth human evaluation carried out by expert psychologists, which was recognized as a key strength by Reviewers WuPB and ArmX.
>
> > **Oracle responses were written by one person and may have subjective biases:**
>
> We agree that having one human writer for oracle responses is a limitation, and this is because of our specific methodology (getting an expert to write a response given the same instructions as the LLM, rather than just taking existing Reddit responses like Ayers et al., 2023), which is more costly (around 15 minutes of expert time per post). But despite this limitation, our results are surprisingly consistent with a lot of work (Ayers et al., 2023; Yin, Jia, and Wakslak, PNAS 2024; Li, Herderich, and Goldenberg, 2024), so we don't think that our results with one human writer should be dismissed out of hand. We'll discuss this issue more in our revision.
>
> > **Performance of LLMs under different constitutions:**
>
> We leave the investigation of LLMs under different prompt words to future work.
>
> > **Other data sources:**
>
> To our knowledge, there is no open-sourced data on cognitive reappraisals that is readily available for access, as such data would be subject to user privacy agreements. For the same reason, we would perform anonymization methods when we release the dataset in the camera-ready version of the paper. As Reviewer LP97 noted as a “key strength” of the paper, the release of our carefully curated dataset would greatly benefit the research community.
>
> **References**
> - Ayers, J. W., Poliak, A., Dredze, M., Leas, E. C., Zhu, Z., Kelley, J. B., Faix, D. J., Goodman, A. M., Longhurst, C. A., Hogarth, M., & Smith, D. M. (2023). Comparing Physician and Artificial Intelligence Chatbot Responses to Patient Questions Posted to a Public Social Media Forum. JAMA internal medicine, 183(6), 589–596.
> - Yin, Y., Jia, N., & Wakslak, C. J. (2024). AI can help people feel heard, but an AI label diminishes this impact. Proceedings of the National Academy of Sciences, 121(14), e2319112121.
> - Li, J. Z., Herderich, A., & Goldenberg, A. (2024). Skill but not Effort Drive GPT Overperformance over Humans in Cognitive Reframing of Negative Scenarios.

---

> > ### Comment · Reviewer_Zhu4 · 2024-06-05
> >
> > I also hope the authors can release the dataset to the research community.

---

### Official Review · Reviewer_LP97 · 2024-05-10

**Rating:** 7
**Confidence:** 4
**Ethics Flag:** 1

**Summary:**

The paper introduces the RESORT framework to guide the large language models (LLMs) to generate reappraisals for emotional support, focusing on the six appraisal dimensions of emotions covering the Reddit posts. Both expert and automatic evaluations are conducted to understand the cognitive appraisals (reappraisals).

**Questions To Authors:**

Other comments and suggestions:

How to determine the order of the six appraisal dimensions in Iterative Guided Refinement?

Carol is a more common name than Erin. Andy, Betty, and Carol also sound harmony.

**Reasons To Accept:**

Pros: the paper may contribute a benchmark dataset to evaluate the large language models (LLMs) in generating (or reframing) reappraisals for emotional support. This contribution subjects to the release of the dataset to the research community.

**Reasons To Reject:**

Cons:

C1: the evaluation samples are limited to 400 posts, 100 for each of the four domains. The oracle responses are 20 plus one highest up-voted comments. This might be relatively small to achieve a reliable evaluation.

C2: the RESORT framework adopts a zero-shot setup to elicit responses. This may be due to the limited number of samples as indicated in the above C1, but in-context learning only requires several examples to prompt a (maybe better) response. Any explorations for these setups?

---

> ### Author Rebuttal · Authors · 2024-05-30
>
> We are grateful for your valuable feedback! We appreciate your recognition of the strength of the RESORT framework in inducing cognitive abilities from LLMs, which Reviewers WuPB and Zhu4 also pointed out. We are also thankful for your appreciation of the systematic and rigorous human evaluation conducted with psychologists holding M.S./Ph.D. degrees, which makes the findings more sound as acknowledged by Reviewers WuPB and ArmX. We address your comments here:
>
> > **Limited number of samples for evaluation:**
>
> We sampled 400 Reddit posts as our source data. This was because our eval method is rigorous and time-consuming, which was acknowledged and appreciated by Reviewers WuPB, Zhu4, and ArmX. The Reddit posts were sourced from 4 different subreddits, namely r/Anxiety, r/Anger, r/Parenting, and r/COVID19 support. We observe a clear difference between the topics in each domain, as shown by the topic modeling results in Table 5 (page 21), and this indicates the linguistic diversity of the samples. In addition, we also highlight the mentally-taxing and cognitively-demanding nature of devising oracle responses. To come up with a single response, the co-author of this study (who is a Ph.D. student in psychology) spent around 15 minutes with undivided attention.
>
> > **Explorations for in-context learning:**
>
> We completely agree that in-context learning setups would be worthwhile to examine LLMs’ ability to provide cognitive reappraisals. We have explored prompting the language models with few-shot oracle examples, but we observe that the model would try to mimic the writing style from the examples, which makes the response more generic. We will include examples from in-context learning in the camera-ready version of the paper. Thank you for the suggestion!
>
> > **Determining the order of the six appraisal dimensions:**
>
> In the current version of the paper, the order of the six appraisal dimensions is set according to the order they were arranged in Table 1. We will explore different orderings and their effects in future work.
>
> > **Change the name in the example to “Carol”:**
>
> Thank you for the suggestion! We will update the name in the example from “Erin” to “Carol” in our camera-ready version of the paper.

---

> > ### Comment · Reviewer_LP97 · 2024-06-05
> >
> > Since the authors have put great efforts into this dataset, I hope it can be released to the research community for follow-up research papers.

---

### Official Review · Reviewer_WuPB · 2024-05-24

**Rating:** 5
**Confidence:** 4
**Ethics Flag:** 1

**Summary:**

The paper presents RESORT, a framework designed to guide large language models (LLMs) in generating cognitive reappraisal responses, a strategy in psychology aimed at changing negative appraisals to improve emotional well-being. The evaluation, conducted by clinical psychologists, demonstrates that even smaller-scale LLMs, when guided by RESORT, can produce effective cognitive reappraisals that significantly outperform human-written responses and non-appraisal-based prompts. The study highlights the potential of LLMs to assist in emotional support through targeted reappraisal, offering a scalable and efficient alternative to traditional methods.

**Questions To Authors:**

Do you have ideas on how to improve the inter-annotator agreement (IAA) for human evaluations, particularly for the ALGN (alignment with reappraisal constitutions) and EMPT (empathy) metrics?

**Reasons To Accept:**

1. Innovative Framework: The paper introduces a novel framework (RESORT) that integrates psychological principles to guide LLMs in generating cognitive reappraisals, addressing a crucial aspect of emotional self-regulation.

2. Expert Evaluation: The evaluation by clinical psychologists provides robust validation of the framework's effectiveness, enhancing the credibility of the findings.

**Reasons To Reject:**

1. Uncomprehensive Evaluation: The evaluation should include comparisons with more large language models such as GPT-40, Gemini, and domain-specific models like MeChat to provide a broader context of the agent’s performance.

2. Closed-Sourced Data: The evaluation is based on closed-sourced data, raising concerns about the reproducibility and transparency of the results. It would be beneficial to conduct evaluations on at least one open-sourced dataset.

3. Annotator Background Clarity: The paper lacks detailed information about the background of the human annotators. Moreover, Table 2 indicates low inter-annotator agreement (IAA) on certain metrics (ALGN and EMPT), which needs to be addressed.

4. Effectiveness of Iterative Guided Refinement: Results in Table 3 suggest that the Iterative Guided Refinement method does not significantly improve performance, questioning its utility in the proposed framework.

---

> ### Author Rebuttal · Authors · 2024-05-30
>
> We are grateful for your positive comments on our innovative RESORT framework that induces cognitive capabilities from LLMs, which Reviewers LP97 and Zhu4 also acknowledged. We are also thankful for your recognition of the in-depth expert evaluation carried out with psychologists holding advanced degrees, which Reviewers Zhu4 and ArmX also appreciated.
>
> > **Uncomprehensive evaluation of LLMs:**
>
> While we agree that it is important to keep up with this rapidly growing field, it should be noted that at the time this paper was submitted (March 2024), GPT-4o was yet to be released and GPT-4 turbo was ranked first on the LMSYS Chatbot Arena Leaderboard. While we agree that evaluating RESORT under different systems would provide a more holistic view of current LLMs’ ability to offer cognitive reappraisals, we highlight that our evaluation was conducted on LLMs at different scales, and our findings suggest that RESORT effectively guides language models at the 7B scale. We also want to point out the rigorous and time-consuming nature of our expert evaluation method, which was appreciated by Reviewers Zhu4 and ArmX.
>
> > **Closed-Sourced Data:**
>
> Ours will be the first open-sourced data of this kind: we will release our annotations and the LLM responses. We will of course first carefully anonymize the responses to preserve user privacy. As for the original posts, we will follow standard practices to provide links to them rather than releasing them.
>
> > **Annotator Background Clarity:**
>
> We recruited 4 expert evaluators from UpWork, who hold either M.S. or Ph.D. degrees in psychology. The details of their background are elaborated in Section 5.2. The advanced expertise of our human evaluators allows for in-depth scrutiny of RESORT, which is acknowledged by Reviewers Zhu4 and ArmX.
>
> > **Effectiveness of Iterative Guided Refinement:**
>
> We highlight that iterative guided refinement is always rated as the best-performing approach, even outperforming oracle responses, which is a key reason to accept this paper suggested by Reviewer ArmX. More importantly, in addition to the improvement in the metric “Alignment with Reappraisal Constitutions”, we also observe a consistent improvement in the level of perceived “Empathy” using iterative guided refinement.
>
> > **Low inter-evaluator agreement on ALGN and EMPT:**
>
> The evaluation task is subjective by nature, although note that for ALGN and EMPT, our expert evaluators already have moderate agreement (Artstein & Poesio, 2008).

---

### Decision · Program_Chairs · 2024-07-10

**Decision:**

Accept

**Comment:**

This paper introduces RESORT, a framework designed to guide large language models (LLMs) in generating cognitive reappraisal responses to improve emotional well-being. Through expert evaluations by clinical psychologists, the study demonstrates that even smaller LLMs, when guided by RESORT, can produce effective cognitive reappraisals that surpass human-written responses and non-appraisal-based prompts. The framework, grounded in cognitive appraisal theories, highlights the potential of LLMs to provide scalable emotional support. Experimental results show LLMs generating reappraisals with high alignment to psychological principles and perceived empathy, offering a promising alternative to traditional methods.
Reviewers appreciates the novelty of the RESORT framework, the comprehensive experiments that have been carried out, and the organized paper writing. There are some cmall concerns around the experimental setup where the groundtruth responses were written by one person and the evaluation set being not large-scale. I think it's understandable for the nature of the study.